# Calcium and Reactive Oxygen Species Signaling Interplays in Cardiac Physiology and Pathologies

**DOI:** 10.3390/antiox12020353

**Published:** 2023-02-02

**Authors:** Bianca De Nicolo, Erica Cataldi-Stagetti, Chiara Diquigiovanni, Elena Bonora

**Affiliations:** 1Department of Medical and Surgical Sciences, University of Bologna, 40138 Bologna, Italy; 2Medical Genetics Unit, IRCCS Azienda Ospedaliero-Universitaria di Bologna, 40138 Bologna, Italy

**Keywords:** calcium, ROS, mitochondria, mitochondrial dysfunction, drugs, cardiomyopathies

## Abstract

Mitochondria are key players in energy production, critical activity for the smooth functioning of energy-demanding organs such as the muscles, brain, and heart. Therefore, dysregulation or alterations in mitochondrial bioenergetics primarily perturb these organs. Within the cell, mitochondria are the major site of reactive oxygen species (ROS) production through the activity of different enzymes since it is one of the organelles with the major availability of oxygen. ROS can act as signaling molecules in a number of different pathways by modulating calcium (Ca^2+^) signaling. Interactions among ROS and calcium signaling can be considered bidirectional, with ROS regulating cellular Ca^2+^ signaling, whereas Ca^2+^ signaling is essential for ROS production. In particular, we will discuss how alterations in the crosstalk between ROS and Ca^2+^ can lead to mitochondrial bioenergetics dysfunctions and the consequent damage to tissues at high energy demand, such as the heart. Changes in Ca^2+^ can induce mitochondrial alterations associated with reduced ATP production and increased production of ROS. These changes in Ca^2+^ levels and ROS generation completely paralyze cardiac contractility. Thus, ROS can hinder the excitation–contraction coupling, inducing arrhythmias, hypertrophy, apoptosis, or necrosis of cardiac cells. These interplays in the cardiovascular system are the focus of this review.

## 1. Introduction

Mitochondria are crucial to maintaining the regulation and performance of different important cellular activities, in particular, with their critical role in energy production for the smooth functioning of energy-demanding organs such as muscles, the brain, and the heart [1]. As a consequence, any dysregulation or alteration in mitochondrial bioenergetics primarily perturbs these organs. Within several cell types and particularly in cardiac myocytes, mitochondria are the major site of reactive oxygen species (ROS) production through the activity of different enzymes (including NADPH oxidase and uncoupled nitric oxide synthase) [2] since they are organelles with great oxygen availability [3].

ROS can act as signaling molecules in a number of different pathways by modulating calcium (Ca^2+^) signaling. Intracellular calcium (Ca^2+^) is the most common second messengers of living cells. This versatility allows it to control diverse processes mediated by rapid Ca^2+^ fluxes, such as contractility, secretion, proliferation, apoptosis, protein folding, and energy metabolism [1,4]. Calcium and ROS mutually influence each other, ROS regulate cellular calcium signaling, whereas calcium signaling is crucial for ROS production [5]. These interplays have been studied in depth in the cardiovascular system and are the focus of this review. In particular, we will discuss how alterations in the crosstalk between ROS and Ca^2+^ can lead to mitochondrial bioenergetics dysfunctions, and the consequent damages for tissues at high energy demand such as heart.

In the heart, maintenance of cellular homeostasis is ensured by a low basal concentration of ROS, as they regulate multiple signaling pathways and physiological processes such as differentiation, proliferation, and excitation–contraction (E-C) coupling [6].

In cardiac cells, mitochondria and the sarcoplasmic reticulum (SR) are closely interconnected, and Ca^2+^ is crucial for optimal function of these two organelles [7]. The controlled release of Ca^2+^ from the SR is necessary for excitation–contraction (E–C) coupling [8]. Mitochondrial dysfunctions are associated with alterations in mitochondrial Ca^2+^ levels [9] and a dysregulated calcium handling is a hallmark in cardiac dysfunction. Ca^2+^ and, ADP, together with the redox state of pyridine nucleotides, actively regulate ROS production in cardiac mitochondria [10]. In fact, a reduction in Ca^2+^ uptake in these organelles and an increased energy demand at the cardiac level induce the oxidation of NADH and NADPH. Thus, the drastically impaired redox potential of the matrix results in increased H_2_O_2_ release [11]. In this context, ROS can hinder the excitation–contraction coupling, inducing arrhythmias, hypertrophy, apoptosis, or necrosis of cardiac cells [12,13]. Finally, in this review we will discuss genes that, when altered, cause cardiomyopathies with mitochondrial dysfunctions.

## 2. ROS

Redox (reduction–oxidation) homeostasis is the dynamic equilibrium of electron transfer reactions, and is related to the concept of free radicals, fundamental to redox signaling and biological function [14]. Free radicals can be oxidants and they are unstable molecular entities with an unpaired electron on the outer layer [14]. As a result of this unpaired electron the free radical undergoes electron transfer reactions, being a reductase when it donates the electron, or an oxidase when it takes an electron from another molecule. The most abundant form of free radicals in the cell originates from diatomic oxygen (O_2_) [15], and they are known as ‘reactive oxygen species’ (ROS).

The term “ROS” does not refer to a specific species, however it is a wide-range term and comprehends different oxygen species with different reactivities and half-lives. The term ROS might refer to both free radicals such as superoxide anion (O_2_•^−^) and hydroxyl radical (•OH), and non-radical oxidants, such as hydrogen peroxide (H_2_O_2_) [16]. Since these molecules are highly reactive, they can react with lipids, proteins, DNA and even other ROS [17].

It is worth noting that the sites of ROS production and Ca^2+^ storage coincide in the cells, at the interface between the plasma membrane and the endoplasmic reticulum (ER) and between this last one and mitochondria.

The major redox signaling agents are the superoxide anion radical (O_2_•^−^) and the hydrogen peroxide (H_2_O_2_). Superoxide is produced mainly through the electron transfer chains (ETC) in mitochondria, but also in other organelles such as the ER and the plasma membrane, by NADPH oxidases (NOX), that catalyze the transfer of an electron from NADPH to O_2_ [18]. In mitochondria, the superoxide radicals (O_2_•^−^) are generated by electrons that escape from complex I and III and that reduce O_2_ [19]. When O_2_•^−^ reacts with another superoxide radical, generates the hydrogen peroxide (H_2_O_2_), which can be reduced to water or partially reduced to hydroxyl radical (•OH), reactions catalyzed by the enzyme superoxide dismutase [20].

Together with O_2_•^−^, H_2_O_2_ is generated in healthy cells at a controlled steady-state level [21] and is physiologically produced by NOXs in the plasma membranes, mitochondria, endoplasmic reticulum, peroxisomes in response to external stimuli [14]. H_2_O_2_ is not a hazardous molecule itself, however it can undergo Fenton reactions through which there is the generation of hydroxyl radical (HO_2_), consequent to the reaction of H_2_O_2_ with the reduced metal ions (Fe^2+^ or Cu^+^). HO_2_ is the most aggressive form of ROS and is the initiator of lipid peroxidation, in fact, can diffuse in lipids, and can produce a carbon-centered radical of polyunsaturated lipids [22].

Like calcium (Ca^2+^) [23], H_2_O_2_ is also a crucial redox signaling agent [14] and the pivotal molecule in homeostatic metabolism, according to the third principle of the Redox Code [24]. In physiological conditions, H_2_O_2_ can oxidize target proteins through reversible reactions—as occur in the reversible oxidation of specific cysteine residues of proteins [25,26,27]—regulating protein activity, localization, and interactions, contributing to organizing cellular processes such as cell proliferation, differentiation, and autophagy [20,28,29].

### The Alpha and Omega of Mitochondrial ROS

Within the cell, the mitochondria are the major site of ROS production, through the activity of different enzymes (complex I and III; oxoglutarate dehydrogenase (OGDH); pyruvate dehydrogenase (PDH); complex II (site IIF) [30]), however, the ETC is the site with the highest production of ROS [31], in particular complexes I and III of the respiratory chain [20,32,33]. In particular, complex I and to a lesser extent complex II release O_2_•^−^/H_2_O_2_ toward the mitochondrial matrix, whereas release from complex III is toward the cristae lumen and the intermembrane space [34,35]. This single electron might reduce oxygen and generate superoxide anion (O_2_•^−^), which is then converted to H_2_O_2_^−^. H_2_O_2_ is generated in the mitochondrial matrix by the action of SOD2 (manganese superoxide dismutase) matrix and in the intermembrane space by SOD1 (Cu, Zn-superoxide dismutase) [3,35,36]. The generated H_2_O_2_ is highly permeable and can be reduced by peroxidases such as glutathione peroxidases (GPx), peroxiredoxins 3 and 5 [37], and catalase (CAT) [38].

Independently of ETC, other mitochondrial enzymes are responsible for ROS production. ROS can be produced in the outer mitochondrial membrane by enzymes such as monoamine oxidase (MAO) and cytochrome b5 reductase (Cb5R) [39], and in the mitochondrial matrix by enzymes of the Krebs cycle such as pyruvate dehydrogenase (PDH) and α-ketoglutarate dehydrogenase (αKGDH), which produce both superoxide and hydrogen peroxide [40,41], and in the inner mitochondrial membrane by glycerol-3-phosphate dehydrogenase and various cytochrome P450 monooxygenases [39]. Interestingly, the activity of the latter enzymes in ROS production is dependent on mitochondrial membrane potential (ΔΨ) [42,43]. Mitochondria produce more ROS at high membrane potential [44]. The closure of the mitochondrial permeability transition pore, the inhibition of complex I and III with rotenone and antimycin A, respectively, and the inhibition of ATP synthase, can all lead to an increase ΔΨ and to an increased ROS production [45,46,47]. However, in pathological conditions, an opposite situation might be also observed, for example, in a neuronal cell model carrying a loss of function *SPART* variant (c.892dupA), cause of Troyer syndrome where a reduced ΔΨ and decreased respiratory activity with a concurrent increase in ROS production have been reported [48].

## 3. Calcium

The crucial role of mitochondria for normal cell physiology is evident in the natural history of various disorders. Indeed, the presence of calcium in mitochondria acts as a double-edged sword: whereas cellular homeostasis is maintained by optimal Ca^2+^ levels, an excess of this ion is reportedly found in many diseases, including neurodegenerative and muscular diseases, such as Huntington’s disease (HD) [49] and Alzheimer’s disease (AD) [50].

Mitochondrial calcium influx is driven by differences in electric changes across the inner mitochondrial membrane and resulting from the proton pumping of the respiratory chain.

The diffusion of Ca^2+^ within the cell is tightly controlled by the elaborate mechanism of cytosolic Ca^2+^ chelation. Under basal conditions, cytosolic calcium concentrations are maintained low and controlled (100 nM) by continuous extrusion to the extracellular environment or uptake by intracellular stores, thus creating a gradient of rapidly increasing Ca^2+^ upon opening of ion pumps and specialized channels [51]. This finely regulated balance allow the genesis of localized Ca^2+^ signals that coordinate the function of target proteins/organs with great spatio-temporal precision [4].

### 3.1. Influx and Efflux of Ca^2+^ in Mitochondria

A variety of targets and Ca^2+^ transport systems are present on the mitochondrial membrane that modulate several mitochondrial functions. The endoplasmic reticulum (ER) represents the primary intracellular Ca^2+^ store and the release of Ca^2+^ occurs through the inositol 1,4,5-trisphosphate receptors (IP3Rs) and ryanodine receptors (RyRs), located in its membranes.

The close proximity and juxtaposition of the ER to mitochondria grants a direct and selective transmission of physiological and pathological Ca^2+^ signals [52]). The membrane contact sites between the endoplasmic reticulum and mitochondria are called mitochondria-associated membranes (MAMs) [53].

The Ca^2+^ transfer between ER and mitochondria through the MAMs depends on a tripartite protein complex that includes IP3R, localized on the ER membrane, voltage-dependent anion channel 1 (VDAC1) residing on the outer mitochondrial membrane (OMM), and the cytosolic glucose-regulated chaperone protein 75 (GRP75) that forms a tether between the two organelles [54].

When the cytosolic Ca^2+^ level is high, the cation is passively transported through the OMM. The presence and function of VDAC1, which enables the transport of all energy metabolites (pyruvate, malate, succinate, NADH, ATP, ADP, and phosphate) from the cytosol to the mitochondria, provides high membrane permeability.

In contrast, the transit of Ca^2+^ across the inner mitochondrial membrane (IMM) is driven by the negative membrane potential and the MCU channel protein [55].

The key components of the MCU channel protein complex include MCU, EMRE, MICU1, and MICU2 (MEMMS) [56]. MCU is the main protein of the holo-complex responsible for the transfer of Ca^2+^ into the mitochondrial matrix. The transmembrane domain of each of the four subunits of MCU (TMD) forms a tetrameric conformation and shapes a pore in the inner mitochondrial membrane effective for Ca^2+^ transfer [57,58,59]. In fact, the TMD contains a ‘DIME’ motif, with conserved amino acids required for MCU-mediated Ca^2+^ uptake: the two residues of aspartic acid and glutamic acid form two parallel side-chain carboxylate rings that act as Ca^2+^ selectivity filter [55,60].

MICU1 forms a large interaction surface area with MCU to seal the intermembrane pore space entrance, while MICU2 binds to MICU1 from the side without contacting MCU [61]. According to literature data, it has been found that MICU1 and MICU2 form a plug to occlude the MCU channel under conditions of low Ca^2+^ concentrations. In the presence of high Ca^2+^ concentrations, these two regulators undergo conformational changes through their EF-handed motif, which result in pore opening and Ca^2+^ permeation into the mitochondria [60]. The conserved aspartate ring of MCU mediates MICU1 binding and regulation in the mitochondrial calcium uniporter complex [60]. EMRE interacts with MICU1 in the intermembrane space and MCU oligomers in the inner membrane. Thus, EMRE appears to operate as a bridge between the channel properties of MCU and Ca^2+^ sensing activity of MICU1/MICU2 [59].

In addition to the Ca^2+^ uptake mechanism, mitochondria present Ca^2+^ release systems, mediated by Na^+^/Ca^2+^ (NCLX, Na^+^/Ca^2+^/Li^+^ exchanger) and H^+^/Ca^2+^ (mHCX, mitochondrial H^+^/Ca^2+^ exchanger) exchangers, which export Ca^2+^ outside the mitochondria. In this way, the organelle limits the accumulation of Ca^2+^ within the matrix and regulates its homeostasis [62].

### 3.2. Key Ca^2+^ Targets and Roles in the Regulation of Mitochondrial Bioenergetics

The controlled uptake of Ca^2+^ in mitochondria regulates the rate of energy production and metabolism, shapes the amplitude and spatio-temporal patterns of intracellular Ca^2+^ signals, and is crucial for programmed cell death [5]. Calcium in mitochondria is critical for the regulation of four dehydrogenases (glycerol phosphate, pyruvate, α-ketoglutarate, and isocitrate dehydrogenase), F0-F1 ATP synthase and two isoforms of the mitochondrial aspartate/glutamate transporter, aralar1 and citrin [56,63]. Of these protein complexes, the two transporters and the glycerol phosphate dehydrogenase have Ca^2+^-binding domains facing the intermembrane space and are affected by changes in the cytoplasmic concentration of calcium ions [56].

In addition, when Ca^2+^ activates the complex F1-F0-ATP synthase, by replacing its natural cofactor (Mg^2+^), the increased steric bulk within the catalytic sites of F1 triggers conformational changes that reverse the function of the complex, and thus ATP synthase hydrolyzes ATP [64].

The other three dehydrogenases are rate-limiting enzymes in feeding electrons at complex I of the ETC [65]. In vertebrates, the mechanisms of activation of these enzymes are all dependent on the accumulation of Ca^2+^ in the mitochondrial matrix [66]. Pyruvate dehydrogenase (PDH) depends on de-phosphorylation of the catalytic subunit by a Ca^2+^-dependent phosphatase [67], while α-ketoglutarate and isocitrate dehydrogenases are activated directly by Ca^2+^ binding [68]. As a result of high mitochondrial Ca^2+^ levels, PDH, α-ketoglutarate, and isocitrate dehydrogenases are activated and stimulate the synthesis of ATP by the mitochondria.

These enzymes are very responsive to changes in Ca^2+^ in the matrix, but the increase in this ion is not the only mechanism that induces their activation; in particular, PDH is also regulated by other allosteric modulators such as pyruvate, ATP, NADH, and matrix pH [66].

Recently, Foskett and collaborators proposed a new regulatory mechanism for cellular bioenergetics, showing that a constitutive reduced Ca^2+^ release through IP3R is crucial for the maintenance of optimal cellular bioenergetics under normal basal conditions because it provides sufficient reducing equivalents to support oxidative phosphorylation [69,70]. In fact, inhibition of IP3R-dependent Ca^2+^ release and, consequently, of mitochondrial Ca^2+^ uptake, causes an overall impairment of cellular bioenergetics. If Ca^2+^ transfer to the mitochondria is absent, an increase in pyruvate dehydrogenase phosphorylation is observed, resulting in its inactivation and decreased TCA (tricarboxylic acid) cycle activity. The slowdown of the TCA cycle reduces NADH and FADH2 production, affecting the activity of the ETC and causing less ATP production. This reduction is detected by the cellular energy sensor AMPK, which, in the presence of a higher AMP/ATP ratio, determines the activation of autophagic processes [70].

Similarly, Filadi et al. described for the first time a new role for TOM70 in modulating ER–mitochondria communication and cellular bioenergetics in mammalian cells [71]. TOM70 is a subunit of the translocase of the outer membrane (TOM) complex and, with the translocase of the inner membrane (TIM), is responsible for the post-translational import of mitochondrial proteins encoded by the nucleus. TOM70 forms clusters along the OMM frequently associated with ER–mitochondria contact sites. Here, it interacts with IP3R isoform 3 and GRP75 (chaperone 75 kDa glucose-regulated protein), stabilizing the functional IP3R-3/GRP75/VDAC1 complex and promoting Ca^2+^ shuttling. This, in turn, promotes and sustains the Krebs cycle and mitochondrial respiration. In fact, the downregulation of TOM70 reduces Ca^2+^ uptake and alters mitochondrial function by reducing ETC activity and ATP synthesis, thereby activating autophagy [71,72].

Regardless of its link to the juxtaposition of ER and mitochondria, the mitochondrial Ca^2+^ uptake process itself plays a crucial role. In 2012, Mallilankaraman and coauthors identified a regulator of the MCU complex (MCUR1), an integral membrane protein required for mitochondrial Ca^2+^ uptake, which was found to be important for maintaining a normal cell metabolism. Knockdown of MCUR1 did not alter MCU localization, but prevented Ca^2+^ uptake by mitochondria. Ablation of MCUR1 also disrupted oxidative phosphorylation, reduced cellular ATP production and oxygen consumption, and finally activated AMP kinase [73].

## 4. The Interplay between Ca^2+^ and ROS

The capacity of mitochondria to accumulate Ca^2+^ is critical for maintaining proper tissue homeostasis. However, Ca^2+^ accumulation in mitochondria leads to decreased ATP production and prolonged opening of the mPTP (permeability transition pore), a high-conductance channel, whose opening allows the release of proapoptotic mitochondrial components [74].

mPTP opening depends not only on Ca^2+^ concentration, but also on other factors including high phosphate concentrations, low adenine nucleotide concentrations and oxidative stress. In fact, Madesh and Hajnoczky showed that O_2_•^−^ is able to induce mPTP opening in a Ca^2+^-dependent manner [75]. The opening of the channel triggers the mitochondrial permeability transition (mPT), which is characterized by a drastic increase in mitochondrial membrane permeability, causing the entry of any molecule with a weight less than 1.5 kDa. This event in turn causes the immediate collapse of the mitochondrial membrane potential (ΔΨ m), membrane depolarization, and ATP depletion. The initial uncoupling effect is followed by the reduction in respiratory activity caused by the loss of pyridine nucleotides and cytochrome c [76]. Swelling of the mitochondrial matrix causes disruption of the outer membrane. Subsequent inhibition of electron flow could explain the increase in ROS formation generated by PTP opening; since the last event is promoted by ROS, a vicious cycle of damage amplification is triggered (Figure 1) [77].

In addition, the continuous release of ROS from mitochondria allows mitochondrial ROS peaks to be maintained during apoptosis. This mechanism may be necessary for signaling to adjacent mitochondria, resulting in global activation of cell death by apoptosis [78].

Proteins, lipids, and nucleic acids can be altered by the accumulation of ROS in mitochondria, which result in covalent modifications and profoundly alter their structure and function [77].

One of the most susceptible targets is cardiolipin, a highly abundant phospholipid in the inner mitochondrial membrane. It has been proposed that oxidation of cardiolipin contributes to complex I impairment [79] and cytochrome c release [80]. Oxidative alterations of mitochondrial lipids and proteins can result in true dysfunction due to alterations in mitochondrial DNA (mtDNA) in the long term. Among the DNA products generated by ROS attack, 8-oxo-deoxyguanosine is the most prevalent [81].

Ben-Kasus Nissim and coworkers have shown that NCLX knockdown increases the mitochondrial Ca^2+^ levels and leads to an stimulates l ROS production in mitochondria [82]. Furthermore, the consequences of specific redox alterations of different isoforms of VDAC are only beginning to be investigated, and it is unclear whether these play a role in VDAC function or might have a role in the pathophysiology of disorders [83].

## 5. Cardiac Muscle, ROS, and Ca^2+^ Signaling

The cardiomyocyte is a cell type that requires high amounts of energy to function efficiently. For these cells, ATP production by oxidative phosphorylation is essential, especially during contraction, after the increase in beating frequency/speed, and for all energy-intensive processes. About one-third of the volume of cardiomyocytes is occupied by mitochondria, which play a crucial role in the efficient coupling between energy production and cell requests [72].

During cardiomyocyte development, the sarcoplasmic reticulum (SR) and mitochondria undergo profound changes in coupling, shape, and distribution. In myoblasts (poorly differentiated muscle cells) mitochondria are few and elongated, much like a reticulum, while in myotubes (differentiated muscle fibers) they appear more spotted and globular. In addition, in myoblasts, mitochondria are coupled to Ca^2+^ release from the ER; in contrast, in myotubes, Ca^2+^ release is driven by SR. in this context, SR is crucial in the regulation of cytoplasmic calcium dynamics and cellular activity.

During differentiation, SR progressively change its shape as a beehive with tubular structure [84]. Ca^2+^ transfer between SR and mitochondria is particularly important in cardiac muscle, where ATP demand is high due to the energy required for excitation–contraction coupling (ECC). Ca^2+^ is uploaded into the mitochondria to invigorate metabolism, generate the ATP needed for contraction, and mediate Ca^2+^ elimination from the cytosol in the relaxation phase [85].

During ECC, the increase in intracytoplasmic calcium required for contraction induction is generated by Ca^2+^-dependent activation of ryanodine receptors (RyRs). These are Ca^2+^ release channels located on the SR, which is the main intracellular storage of this ion. In cardiomyocytes both RyRs and inositol 1,4,5-trisphosphate receptors (IP3Rs) are present. In myoblast mitochondria, Ca^2+^ release from the ER is mediated by IP3R. During cell differentiation, the expression of RyRs (particularly isoform 2) increases dramatically in cardiac muscle, and this is critical for excitation–contraction coupling. Simultaneously, IP3Rs play different roles, such as regulating gene transcription and hypertrophy [84,86], although the physiological contribution of IP3R-mediated Ca^2+^ release is not completely elucidated [85].

As it is the case in other cell types, also in cardiomyocytes the transfer of Ca^2+^ from SR to mitochondria involving IP3Rs in MAMs is mediated by the VDAC1-GRP75 (a mammalian heat shock stress protein 70). In this context, IP3R1 is called up, binds VDAC1 through GRP75 that tethers the two proteins by binding to their regions exposed in the cytosol and therefore forming a channel complex through which there is Ca^2+^ transfer between SR and mitochondria, [87].

### 5.1. Calcium and ROS in Heart Failure

Mitochondria and SR are interrelated and connected, and Ca^2+^ is crucial for the functioning of both organelles. Mitochondrial dysfunction is associated with alterations in Ca^2+^ levels [87,88,89,90,91,92,93,94,95].

Dysregulated calcium handling is a hallmark in cardiac dysfunction. In particular, changes in Ca^2+^ can induce mitochondrial alterations with reduced ATP production and increased production of ROS. Thus, ROS can hinder the excitation–contraction coupling, inducing arrhythmias, hypertrophy, apoptosis, or necrosis of cardiac cells [96,97,98,99,100,101]. Different studies reported a correlation between an increased level of ROS (in plasma and heart) and the gravity of left ventricular dysfunction [102,103,104].

Heart failure induces the activation of a number of mechanisms that result in a reduction in mitochondrial calcium transfer. Specifically, there is an increase Ca^2+^ influx in through the NCLX due to increased Na^+^ in the cytosol, reduction in MCU opening, SR Ca^2+^-ATPase activity, and ryanodine receptor expression. The final combined effect of these events is the negative alteration of the Krebs cycle [105,106,107,108,109,110]. As a result, the accumulation of oxidized pyridine nucleotides prevents the production of ATP from NADH and the detoxification of ROS by NADPH. The antioxidative enzyme responsible for the regeneration of NADPH from NADH is the nicotinamide nucleotide transhydrogenase (NNT) [111]: under pathological conditions, when the metabolic demand increases, the direction of the NNT reaction is reversed, oxidizing NADPH to regenerate NADH and produce ATP, and interfering with the NADPH-dependent antioxidative capacity [10]. Because mitochondria represent the main ROS scavenging system of the cell [103], the oxidation of NADPH by NNT could cause excessive mitochondrial ROS release, that leads to necrosis, left ventricular dysfunction, and death [112].

### 5.2. Mitochondrial ROS vs. ER ROS

Mitochondria have been considered the primary source of ROS [7,113,114,115,116,117,118], but several lines of evidence propose an important role also for ROS generated in the ER in cardiovascular diseases.

MAMs are the contact sites between the endoplasmic reticulum or sarcoplasmic reticulum. Our knowledge of calcium signaling in cardiac pathologies, where ER and oxidative stresses are predominant [5,119,120,121], suggests that calcium may, in fact, be the cause, rather than the effect, of altered mitochondrial ROS. This implies that calcium overload signals mitochondria to produce lethal levels of ROS.

The ER is a primary site of protein synthesis and post-translational processing, and the protein folding process is influenced by the redox state of the ER. A well-known condition of the ER, called ER stress, due to the accumulation of misfolded polypeptides, is caused by the accumulation of oxidant equivalents in the ER. ER stress activates the unfolded protein response (UPR), which raises the folding protein capacity, resulting in an increased production of oxidative equivalents and an additional deterioration of the redox state [122]. During ER stress calcium channels, both ryanodine and inositol-3-phosphate receptors open [123] with the release of calcium, which is crucial for the contraction of the muscles. To sustain Ca^2+^ homeostasis, Ca^2+^ returns to the SR/ER through SERCA (sarco-endoplasmic reticulum calcium ATPase). In pathological conditions, Ca^2+^ homeostasis in the ER is dysregulated, with an enhanced Ca^2+^ release from mitochondria [124,125]. Ca^2+^ within the mitochondria generates superoxide, a marker for oxidative stress [126]. Cardiac contractility is entirely paralyzed by Ca^2+^ leakage, Ca^2+^ overload, and ROS generation.

### 5.3. In Vivo Mouse Model of Postmyocardial Infarction

Type 2 ryanodine receptor (RyR2) and type 2 inositol 1,4,5-trisphosphate receptor (IP3R2) [127,128,129] are the two intracellular Ca^2+^ channels in SR of cardiac cells, and the RyR2 receptor is crucial for cardiac excitation–contraction (E–C) coupling in cardiomyocytes.

In several murine models of postmyocardial infarction that have been generated so far, it has been shown that, based on the physical and functional association between the SR and mitochondria, when the SR calcium was released in cardiomyocytes by the two main channels, it accumulated in mitochondria, bringing to mitochondrial dysfunction, oxidative stress and decreased ATP production [88,130,131].

To investigate whether Ca^2+^ accumulation in mitochondria of failing hearts was caused by Ca^2+^ leak through the RyR2 receptor, Santulli et al. generated a murine model carrying a homozygous RyR2 mutation that renders the channel leaky (RyR2^S2808D/S2808D^) and a mouse model with a homozygous RyR2 mutation that renders the channel protected against leak (RyR2^S2808A/S2808A^) [129]. In cardiomyocytes derived from the RyR2^S2808D/S2808D^ mice, there was an increased mitochondrial Ca^2+^ and ROS production, the presence of dysmorphic and malfunctioning mitochondria, a reduction in mitochondrial size, and low fusion-to-fission ratio compared with wild-type (WT) and RyR2^S2808A/S2808A^ cardiomyocytes [129].

In opposition, cardiac-specific deletion of the IP3R2 receptor did not produce consequences on mitochondrial Ca^2+^ accumulation, as observed in the murine model (IP3R2^CVKO^), in which IP3R2 expression was suppressed in ventricular cardiomyocytes. IP3R2^CVKO^ mice survived to adulthood without showing alteration in mitochondrial function. Ca^2+^ sparks, SR Ca^2+^ load, mitochondrial Ca^2+^ level, and ROS production were not affected in IP3R2^CVKO^ ventricular cardiomyocytes. Moreover, IP3R2^CVKO^ mice exhibited normal myocardial mitochondria and no significant effect after postmyocardial infarction [129].

### 5.4. Drug Targeting of Mitochondrial Ca^2+^ and Homeostasis

Based on the observed experimental data, the mitochondrial redox state might be a plausible drug target in cardiac alterations. Two distinct approaches have been used to modulate ROS and Ca^2+^ in mitochondria: (i) direct targeting of mitochondrial ROS by agents that accumulate in mitochondria; (ii) normalization of mitochondrial Ca^2+^ signaling to guarantee a balance between mitochondrial ROS emission and detoxification.

For the direct targeting of mitochondrial ROS, drugs that have been developed to accumulate in mitochondria or scavenge ROS (MitoQ) or directly act at the ETC level to reduce ROS production (SS-31). As an example, CGP-37157 is a benzodiazepine compound selectively inhibiting NCLX, the mitochondrial Na^+^/Ca^2+^ exchanger by which mitochondrial Ca^2+^ is extruded. CGP-37157 suppresses mitochondrial Ca^2+^ efflux, abolishes NADH oxidation, and reduces ROS production in failing cardiomyocytes [110]. Overall, experimental studies performed with CGP-37157 showed the positive effects of mitochondrial NCLX inhibition in animal models of heart failure and in isolated heart cells. However, currently, there are no clinical trials to test the effect of these drugs.

A different strategy to normalize mitochondrial Ca^2+^ signaling is to reduce the concentration of intracellular Na^+^. In heart failure, [Na^+^]i is increased [132], and during excitation–contraction coupling, it hinders mitochondrial Ca^2+^ accumulation since it increases mitochondrial Ca^2+^ efflux via the NCLX [102]. It has been shown that an elevated late Na^+^ current (late INa) is required to increase cytosolic Na^+^. An enhanced late INa has been related to elevated mitochondrial ROS emission. This, in turn, worsens late INa through Ca^2+^-/calmodulin-dependent kinase II (CaMKII) oxidation. CaMKII [133,134] can enhance late Na^+^ current and cause arrhythmias during HF [135].

Ranolazine is a piperazine derivative inhibitor of late Na^+^ current [136] that was approved by the FDA in 2006 for chronic angina and cardiac ventricular dysrhythmias [137]. MERLIN-TIMI 36 trials provided the hypothesis that ranolazine may be particularly beneficial in patients with HF [138]. In this trial, patients with acute coronary syndrome profited from ranolazine, particularly when they had elevated levels of brain natriuretic peptide as a marker of HF [138]. Evidence suggested that the mechanisms of action could be the blocking of the late Na^+^ current that occurs during ischemia, the blocking of mitochondrial complex I activity, or by modulating mitochondrial metabolism [139]. In mitochondria isolated from the hearts of patients treated with ranolazine, there was decreased cytochrome c release and mild resistance to the opening of mPTP when compared with control hearts. Through its late Na^+^ current blocking action, ranolazine protected the heart during IR injury by decreasing the load of cytosolic (c)Ca^2+^ and mitochondrial (m)Ca^2+^. The final effect was a reduction in necrosis and apoptosis [139].

Recently, empagliflozin has demonstrated its positive effects in regulating cytosolic Na^+^ and mitochondrial Ca^2+^. It selectively inhibits the sodium glucose cotransporter 2 channel (SGLT-2) [140], which reabsorbs the filtered glucose in the renal early proximal tubule. Its main role is to reabsorb the majority of filtered glucose, decreasing renal glucose reabsorption and, thus, hyperglycemia. A clinical trial in 7020 patients, the EMPA-REG OUTCOME study, showed that patients with type 2 diabetes at high cardiovascular risk and treated with empagliflozin showed ameliorated cardiovascular endpoints and reduced death from HF when the study drug was added to standard care [141,142]. Empagliflozin might reduce [Na^+^]i in cardiac myocytes independently of SGLT inhibition, possibly by interacting with the Na^+^/H^+^ exchanger [143]. Thereby, this drug empowered mitochondrial Ca^2+^ in cardiac myocytes.

Antioxidants such as vitamins E and C have been used in animal experiments, but promising results were observed in preclinical studies only [144]. In subsequent clinical trials, none of these compounds led to any significant benefit. Despite evidence that mitochondria are crucial for oxidative damage, compartmentalization and the interaction between compartments probably explain why “generic” antioxidants, such as vitamins, failed to target dysfunctional mitochondria and altered Ca^2+^ levels.

A more targeted and specific approach has been focused on NOXs, which have the biological role of producing ROS for signaling purposes and transferring electrons across biological membranes to produce O_2_^−^ [145]. Along with the already well-known lipid-lowering effect of statins in preventing cardiovascular disease, their antioxidant effect through NOX2 inhibition has been demonstrated. Statin treatment provided positive antioxidant effects in vitro in isolated cardiomyocytes and in animal models of HF [141,142]. However, in the CORONA study, comprising 5011 HF patients, statin therapy did not show to reduce cardiovascular death [146].

In January 2021, the FDA approved vericiguat, a new soluble oral guanylate cyclase stimulator, which enhances the production of cyclic guanosine monophosphate, for the treatment of chronic HF. It is well known that the cardioprotective pathway of soluble nitric oxide guanylate cyclase-cyclic guanosine monophosphate is compromised in patients with HF [147].

Based on data from the SOCRATES-REDUCED and VICTORIA trials in adult patients with chronic HF, this drug was approved for medical treatment. In the VICTORIA trial, the dose of vericiguat was initially 2.5 mg/day and increased after 2 weeks to 5 mg/day and then to 10 mg/day. Placebo doses were administered in the same manner. After about 1 year, 90% of patients were treated with the 10 mg target dose. The median follow-up for the primary endpoint was 11 months. The annualized absolute risk reduction with the drug treatment was ~4.2% over the course of the study [148]. The recommended initial oral dose of vericiguat was 2.5 mg/day, with food intake, and should be doubled approximately every 2 weeks to reach the target maintenance dose of 10 mg/day. Based on the VICTORIA trial, vericiguat significantly reduced the rate of hospitalization and cardiovascular death attributed to HF. In fact, regardless of patients’ atrial fibrillation status, vericiguat was better in preventing all causes of death, cardiovascular death, hospitalization, and HF.

The main advantage of this drug is that it avoids the risk of electrolyte imbalance or renal damage [149]. Moreover, vericiguat bypasses the many problems of current therapies, such as the gradual decline in effectiveness, drug dose-dependent tolerance, and off-target effects due to a lack of specificity, thus representing a game changer in the treatment of HF, because it has great potential to reduce the severity of the disease [150].

### 5.5. Cardiomyopathy with Mitochondrial Dysfunction-Associated Genes

Several signaling pathways, ion homeostasis, and metabolism are impaired in cardiomyopathies. Therefore, mitochondria, which are the cellular powerhouse, are involved in many of these processes. The known genes that are altered in cardiomyopathies with mitochondrial dysfunctions are reported in Table 1.

#### 5.5.1. *MTO1*

*MTO1* encodes for a mitochondrial protein involved in tRNA modification and protein synthesis and catalyzes the 5-carboxymethyl aminomethylation of the uridine base in mitochondrial tRNAs that transport Gln, Glu, and Lys. Mutations in this gene cause an autosomal recessive disorder known as combined oxidative phosphorylation deficiency 10 (COXPD10), characterized by altered OXPHOS activity. Alterations in mitochondrial oxidative respiration cause hypertrophic cardiomyopathy and lactic acidosis in early infancy and complications that can be fatal in severe cases [151].

#### 5.5.2. *AGK*

The human *AGK* gene encodes for the mitochondrial acylglycerol kinase, an enzyme located on the mitochondrial membrane involved in the formation of phosphatidic and lysophosphatidic acids. Homozygous or compound heterozygous mutations in the *AGK* gene have been associated with Sengers syndrome, also known as cardiomyopathic mitochondrial DNA (mtDNA) depletion syndrome-10 (MTDPS10). This syndrome is characterized by skeletal myopathy, hypertrophic cardiomyopathy, exercise intolerance, lactic acidosis, and congenital cataracts. Skeletal muscle biopsies showed severe mtDNA depletion [152], and cardiomyopathy is the major cause of early death [157].

#### 5.5.3. *SLC25A4*

*SLC25A4* gene encodes for the Solute Carrier Family 25 Member 4, expressed in mitochondria. The function of this protein is to translocate ADP from the cytoplasm into the mitochondrial matrix and ATP in the opposite direction. Homozygous or compound heterozygous mutations in this gene cause mitochondrial DNA depletion syndrome 12B (MTDPS12B), an autosomal recessive mitochondrial disease. Onset occurs in infancy with progressive hypertrophic cardio- and skeletal myopathy. The presence of red and irregular fibers, accumulation of abnormal mitochondria, and mtDNA depletion can be observed in skeletal muscle biopsies from the affected individuals [153].

#### 5.5.4. *MT-TL1*

*MT-TL1* is a mitochondrial gene that encodes the t-RNA-Leu. Mutations in *MT-TL1* can result in multiple mitochondrial deficiencies and associated disorders. Among them, variants in *MT-TL1* cause hypertrophic cardiomyopathy with renal abnormalities due to altered oxidative phosphorylation. This syndrome is characterized by hypertrophic and dilated cardiomyopathy, myopathy with hypotonia with developmental delay, and/or regression with cerebral atrophy and chronic renal [154].

#### 5.5.5. *MT-TK*

Another mitochondrial RNA gene is *MT-TK*, affiliated with the tRNA class, which transfers the amino acid lysine during translation. Defects in this gene are associated with mtDNA-related cardiomyopathy and hearing loss. The clinical manifestation of this rare mitochondrial disease is characterized by progressive sensorineural hearing loss along with hypertrophic cardiomyopathy and encephalomyopathy. Other symptoms might be present such as progressive external ophthalmoparesis (PEO), ataxia, slowed speech, myalgia, and muscle weakness [155].

#### 5.5.6. *TAFAZZIN*

The *TAFAZZIN* gene encodes a protein expressed at high levels in cardiac and skeletal muscle. Barth syndrome is caused by variants in the *TAFAZZIN* gene [87,88,89,90,91,92,93,94,95,154]. This syndrome is an X-linked disease with dilated cardiomyopathy (CMD) with endocardial fibroelastosis (EFE), a proximal skeletal myopathy, growth retardation, neutropenia, and organic aciduria [156]. Hypertrophic cardiomyopathy, ventricular arrhythmia, motor delay, and other cardiac and motor symptoms might be present.

## 6. Conclusions and Future Perspectives

In cardiac cells, ATP production by oxidative phosphorylation is essential, especially during contraction, after the increase in beating frequency/speed, and for all energy-intensive processes. As a consequence, dysregulation in mitochondrial bioenergetics severely affects their function. Changes in Ca^2+^ can induce mitochondrial alterations associated with reduced ATP production and increased production of ROS. In this context, ROS can hinder the excitation–contraction coupling, inducing arrhythmias, hypertrophy, apoptosis, or necrosis of cardiac cells [97,98,99,100,101]. The search for new methods to recover these damages has seen a great expansion in recent years through the development of functional biomaterials and biomimetic scaffolds in regenerative medicine, which finds fertile ground for the application of cardiac therapies, considering the low regenerative capacity of this tissue [158].

Many efforts have been made in order to develop functional cardiac tissue in vitro, especially with the development of knowledge in the field of pluripotent stem cells (PSCs). Human pluripotent stem cells (hPSCs) have the capacity to differentiate into different cell types. They can be derived from the cells within the blastocyst at the beginning of embryonic development, from which the various structures of the fetus originate [159], or they can be derived from human cells that are de-differentiated by expression of Yamanaka factors (Oct3/4, Sox2, Klf4, c-Myc), known as induced pluripotent stem cells (iPSCs) [160].

The use of iPSCs is an emerging application for the generation of disease models and the search for new therapies, making it possible not to use animal models and to overcome the ethical problems associated with human embryonic cells. The reprogramming of cells takes place in the context of the individual’s genetic background, and it has been shown how the genetic and epigenetic behaviors of iPSCs reflect those of the donor individual’s cells. Therefore, iPSCs have the potential to generate tissue-like cells or structures, such as cardiomyocytes, that are exactly those of the patient with the disease of interest, including the main inherited cardiomyopathies that we previously cited.

Branco et al. recently developed a protocol for the generation of self-organized human multilineage organoids that recreate the co-presence of several specialized cells [161]. Co-culturing these organoids together with cardiomyocytes, a functional cardiac organoid surrounded by myocardial-like tissue was generated, recapitulating the structure and function of mature cardiac cells [161].

Tenreiro et al. reported data from trials aimed at generating cardiomyocytes from cells derived from cardiomyopathy patients and studying their molecular mechanisms [162]. An important example of this approach is DMDstem (NCT03696628), a trial completed in 2021 and performed in children with genetic cardiomyopathy vs. healthy ones. However, the outcomes of this study are not yet reported in the literature. Another trial, IndivuHeart (NCT02417311), initiated in 2014 and completed in 2019, aims to ensure individualized early risk assessment for heart disease and to make engineered heart tissue technology (hiPSC-EHT) a clinically applicable strategy [163].

Advances in hiPSCs biology, genome-editing technologies, and cardiac tissue engineering all concur to generate healthy cardiac tissue from the patient himself, opening doors for targeted therapy and personalized medicine.

## Figures and Tables

**Figure 1 antioxidants-12-00353-f001:**
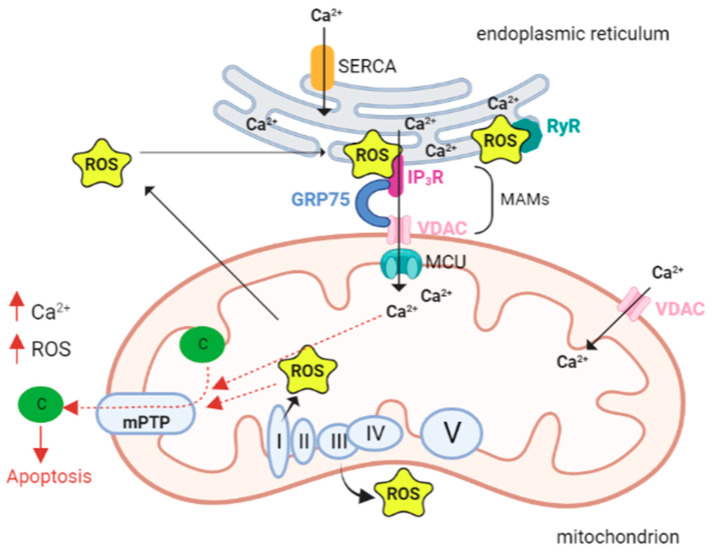
Calcium and ROS interplay in mitochondria. The close proximity and juxtaposition of the ER to mitochondria facilitate a straight and selective transport of calcium (Ca^2+^). The mitochondria-associated membranes (MAMs) are the site of contact between the endoplasmic reticulum (ER) and mitochondria. Here calcium release channels accumulate, and these channels include the IP3R localized on the ER membrane, the voltage-dependent anion channel 1 (VDAC1) on the outer mitochondrial membrane (OMM), and the cytosolic glucose-regulated chaperone protein 75 (GRP75) that forms a tether between the two organelles. Increased levels of mitochondrial calcium stimulate the activity of the electron transport chain leading to a higher release of reactive oxygen species (ROS). As a vicious circle ROS can hit Ca^2+^ channels localized in the ER membranes and cause a further leak of Ca^2+^ from the ER that leads to increased ROS production in the mitochondria. A pick in the levels of both Ca^2+^ and ROS opens the mitochondrial permeability transition pore (mPTP), allowing the release of cytochrome c that leads to the activation of apoptosis. Abbreviations: ROS reactive oxygen species; Ca^2+^ calcium ion, IP3R IP3 receptor; GRP75 glucose-regulated chaperone protein 75; VDAC voltage-dependent anion channel; MCU mitochondrial calcium uniporter; c cytochrome c; SERCA sarco/endoplasmic reticulum Ca^2+^ ATPase; RyR ryanodine receptors; mPTP—mitochondrial permeability transition pore; I, II, III, VI, V respiratory complex I–V. Created with BioRender.com.

**Table 1 antioxidants-12-00353-t001:** Susceptibility loci and candidate genes linked to cardiomyopathies with mitochondrial alterations.

Chromosome Location	Phenotype	Phenotype MIM Number	Inheritance	Gene/Locus	Gene/Locus MIM Number	References
6q13	Combined oxidative phosphorylation deficiency 10	614702	AR	*MTO1*	614667	[151]
7q34	Sengers syndrome	212350	AR	*AGK*	610345	[152]
4q35.1	Mitochondrial DNA depletion syndrome 12B (cardiomyopathic type)	615418	AR	*SLC25A4*	103220	[153]
Mt	Hypertrophic cardiomyopathy with kidney anomalies due to mtDNA mutations		Mitochondrial inheritance	*MT-TL1*		[154]
mt	Mitochondrial DNA- related cardiomyopathy and hearing loss		Mitochondrial inheritance	*MT-TK*		[155]
Xq28	Barth syndrome	302060	XLR	*TAFAZZIN*	300394	[156]

## Data Availability

Not applicable.

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
