# Peer review of "Calcium and Reactive Oxygen Species Signaling Interplays in Cardiac Physiology and Pathologies"

_antioxidants, 2023, doi:10.3390/antiox12020353_

Round 1

Reviewer 1 Report

The review of De Nicolo and coll. offers a complete picture about pathophysiological aspects related to calcium and oxidative stress. 

Only minor comments:

- The part relative to the therapeutical approaches should better underline as, in front of the amount of evidence demonstrating the pathophysiological relevance of calcium and ROS interplays, there are very few effective therapeutical approach.

- The authors report ranolazine as possible therapeutic strategy but the evidence supporting its usefulness are weak.

- Concerning SGLT2 inhibitors, the authors should describe more in depth the potential but controversial effects at the level con cardiomyocite (e.g. inhibition of Na/H exchanger).

- Also the recent evidence about vericiguat in heart failure should be discussed.

Author Response

Response to Reviewer 1 Comments

Point 1. The part relative to the therapeutical approaches should better underline as, in front of the amount of evidence demonstrating the pathophysiological relevance of calcium and ROS interplays, there are very few effective therapeutical approach.

We agree with the reviewer's comments and modified the text accordingly on the therapeutic approaches used to treat cardiomyopathies:

Page 10, lines 445-451: “As an example, CGP-37157 is a benzodiazepine compound selectively inhibiting NCLX, the mitochondrial Na+/ Ca2+ exchanger by which mitochondrial Ca2+ is extruded. CGP-37157 suppresses mitochondrial Ca2+ efflux, abolishes NADH oxidation and reduces ROS production in failing cardiomyocytes [110]. Overall, experimental studies performed with CGP-37157 showed the positive effects of mitochondrial NCLX inhibition in animal models of heart failure and in isolated heart cells. However, currently there are no clinical trials to test the effect of these drugs”.

Page 10, lines 453-458: “In heart failure, [Na+]i is increased [132] and during excitation-contraction coupling, it hinders mitochondrial Ca2+ accumulation, since it increases mitochondrial Ca2+ efflux via the NCLX [102]. It has been shown that an elevated late Na+ current (late INa) is required to increase cytosolic Na+. An enhanced late INa has been related to elevated mitochondrial ROS emission. This in turn worsens late INa through Ca2+-/calmodulin-dependent kinase II (CaMKII) oxidation”.

Page 11, lines 486-497: “Despite evidence that mitochondria are crucial for oxidative damage, compartmentalization and the interaction between compartments probably explain why "generic" antioxidants, such as vitamins, failed to target dysfunctional mitochondria and altered Ca2+ levels.

A more targeted and specific approach has been focused on NOXs, which have the biological role to produce ROS for signaling purposes and to transfer electrons across biological membranes to produce O2- [143]. Along with the already well-known lipid-lowering effect of statins in preventing cardiovascular disease, their antioxidant effect through NOX2 inhibition has been demonstrated. Statin treatment provided positive antioxidant effects in vitro in isolated cardiomyocytes and in animal models of HF [144,145]. However, in the CORONA study, comprising 5011 HF patients, statin therapy did not show to reduce cardiovascular death [146]”.

Point 2. The authors report ranolazine as possible therapeutic strategy but the evidence supporting its usefulness are weak.

We thank the reviewer for this important suggestion and modified the text as follows:

Page 10, lines 463-472: “In this trial, patients with acute coronary syndrome profited from Ranolazine, particularly when they had elevated levels of brain natriuretic peptide as a marker of HF [138]. Evidence suggested that the mechanisms of action could be the blocking of the late Na+ current that occurs during ischemia, the blocking mitochondrial complex I activity, or by modulating mitochondrial metabolism [139]. In mitochondria isolated from hearts of patients treated with Ranolazine, there was decreased cytochrome c release and mild resistance to the opening of mPTP, when compared with control hearts. Through its late Na+ current blocking action, Ranolazine protected the heart during IR injury, by decreasing the load of cytosolic (c)Ca2+ and mitochondrial (m)Ca2+. The final effect was a reduction in necrosis and apoptosis [139].”.

Point 3. Concerning SGLT2 inhibitors, the authors should describe more in depth the potential but controversial effects at the level con cardiomyocite (e.g. inhibition of Na/H exchanger).

We modified the text in order to provide more details on this point:

Page 10-11, lines 473-483: “Recently, Empagliflozin has demonstrated its positive effects in regulating cytosolic Na++ and mitochondrial Ca2+. It selectively inhibits the sodium glucose cotransporter 2 channel (SGLT-2) [140], that reabsorbs the filtered glucose in the renal early proximal tubule. Its main role is to reabsorbs the majority of filtered glucose, decreasing renal glucose reabsorption and thus hyperglycemia. A clinical trial in 7020 patients, the EMPA-REG OUTCOME study, showed that patients with type 2 diabetes at high cardiovascular risk and treated with Empagliflozin, showed an ameliorated cardiovascular endpoints and reduced death from HF, when the study drug was added to standard care [135,136,137]. Empagliflozin might reduce [Na+]i in cardiac myocytes independently of SGLT inhibition, possibly by interacting with the Na+/H+ exchanger [141]. Thereby, this drug empowered mitochondrial Ca2+ in cardiac myocytes”.

Point 4. Also the recent evidence about vericiguat in heart failure should be discussed.

This is another great suggestion. Accordingly, we added insight on this newly approved drug, which seems to be more effective than current therapies.

Page 11, lines 498-520: “In January 2021, FDA has approved Vericiguat, a new soluble oral guanylate cyclase stimulator, which enhances the production of cyclic guanosine monophosphate, for the treatment of chronic HF. It is well known that the cardioprotective pathway of soluble nitric oxide guanylate cyclase-cyclic guanosine monophosphate is compromised in patients with HF [148].

Based on data from the SOCRATES-REDUCED and VICTORIA trials in adult patients with chronic HF, this drug was approved for medical treatment. In the VICTORIA trial, the dose of Vericiguat was initially 2.5 mg/day and increased after 2-weeks to 5 mg/day, and then to 10 mg/day. Placebo doses were administered in the same manner. After about 1 year, 90% of patients were treated with the 10 mg target dose. The median follow-up for the primary endpoint was 11 months. The annualized absolute risk reduction with the drug treatment was ~4.2% over the course of the study [147]. The recommended initial oral dose of Vericiguat was 2.5 mg/day, with food intake, and should be doubled approximately every 2 weeks to reach the target maintenance dose of 10 mg/day. Based on the VICTORIA trial, Vericiguat significantly reduced the rate of hospitalization and cardiovascular death attributed to HF. In fact, regardless of patients' atrial fibrillation status, Vericiguat was better in preventing all causes of death, cardiovascular death, hospitalization and HF.

The main advantage of this drug is that it avoids the risk of electrolyte imbalance or renal damage [149]. Moreover, Vericiguat bypasses the many problems of current therapies, such as the gradual decline in effectiveness, drug dose-dependent tolerance, and off-target effects due to a lack of specificity, thus representing a game changer in the treatment of HF, because it has great potential to reduce the severity of the disease [150]”.

We thank the reviewer for the helpful suggestions and comments of our manuscript.

Reviewer 2 Report

The article focused calcium and ROS interation, esp. in mitrochordia, with mainly emphasizing dysregulation in cardiovascular system. The review is comprehensive, extensive, and carried new information for further disease management. Overall, the article merits publication in antioxidants. 

Author Response

We sincerely thank the reviewer for the positive comment on the manuscript. We are satisfied that our work on disease management information gathering was appreciated.

Reviewer 3 Report

This review might be interesting for potential readers after extensive revision to improve logical flow, readability, terminology, and the relation of the content to the review focus and title. As it is currently written, this review is hard to follow, especially with respect to cardiac dysfunction.

Several examples of necessary improvements (but not limited to) are summarized below.

The current review title, “Cardiac dysfunction, Calcium signaling and reactive oxygen species interplays” is misleading and does not reflect the major contest of the review. The more appropriate title could be “Cardiac calcium signaling and reactive oxygen species interplays” or “Calcium and reactive oxygen species signaling interplays in cardiac physiology and pathologies.”

Introduction: 1st paraph of the Introduction should introduce the importance of ROS and Ca signaling in the heart rather than mitochondria.

Presented data need to be better organized as cell-based, experimental animal models and clinical.

Please fix the used terminology. Thus, “failing heart” is not equal to cardiomyopathies. Heart failure needs to be clearly defined and used appropriately, as “cardiac dysfunction” and “post-myocardial infarction” are not equal to “heart failure.”

Several review parts are superficially written and need more specific ROS/Ca-signaling-related details. For instance:

In subsection - 5.4 Drug targeting of mitochondrial Ca2+ and homeostasis:

“Ranolazine is a piperazine derivative selectively inhibiting late INa [132], 2007). It was FDA approved in 2006 for chronic angina [133]. MERLIN-TIMI 36 trials provided us the hypothesis that ranolazine may be particularly beneficial in patients with HF [134].

Empagliflozin is a selective inhibitor of the sodium glucose cotransporter 2 (SGLT-2) [135] that is located in the renal early proximal tubule and reabsorbs the majority of filtered glucose.

Empagliflozin decreases renal glucose reabsorption and thus hyperglycaemia [136,137]. A trial in 7020 patients called EMPA-REG OUTCOME showed that empagliflozin reduced several cardiovascular endpoints and in particular, hospitalization for and death from heart failure [138]. “

6. Conclusions and Future Perspectives:

The many statements are unrelated to the current review and should be removed.

The following statement is misleading - “Interestingly, clinical trials are already underway whose aim is to generate differentiated cardiac cells from cells derived from cardiomyopathy patients and study their molecular mechanisms [153].” Unclear the type of cardiac cells and the clinical trial(s) the authors refer to.

Author Response

Response to Reviewer 1 Comments

Point 1. The current review title, “Cardiac dysfunction, Calcium signaling and reactive oxygen species interplays” is misleading and does not reflect the major contest of the review. The more appropriate title could be “Cardiac calcium signaling and reactive oxygen species interplays” or “Calcium and reactive oxygen species signaling interplays in cardiac physiology and pathologies.”

Page 1, lines 2-3: “Calcium and reactive oxygen species signaling interplays in cardiac physiology and pathologies”.

We thank the reviewer for the suggestion and agree that the new title is more suitable.

Point 2. Introduction: 1st paraph of the Introduction should introduce the importance of ROS and Ca signaling in the heart rather than mitochondria.

We have modified accordingly to the reviewer’s suggestions:

Page 1, lines 34-35: “Within several cell types and particularly in cardiac myocytes”.

Page 1, lines 36-37: “including NADPH oxidase and uncoupled nitric oxide synthase”.

Page 2, lines 48-51: “In the heart, maintenance of cellular homeostasis is ensured by a low basal concentration of ROS, as they regulate multiple signaling pathways and physiological processes such as differentiation, proliferation, and excitation-contraction (E-C) coupling [6]”.

Page 2, lines 57-61: “Ca2+ and, ADP, together with the redox state of pyridine nucleotides, actively regulate ROS production in cardiac mitochondria [10]. In fact, a reduction in Ca2+ uptake in these organelles and an increased energy demand at the cardiac level induce the oxidation of NADH and NADPH. Thus, the drastically impaired redox potential of the matrix results in increased H2O2 release [11]”.

Point 3. Presented data need to be better organized as cell-based, experimental animal models and clinical.

We carefully revised and modified the text to improve our manuscript. All changes have been highlighted in red.

Regarding the experimental animal models of postmyocardial infarction, it was specified that only homozygous mice show the mutant phenotype. It was also clarified that these are two examples of mouse models among the many that exist:

Page 9-10, lines 419-427: “To investigate whether Ca2+ accumulation in mitochondria of failing hearts was caused by Ca2+ leak through the RyR2 receptor, Santulli et al. generated a murine model carrying a homozygous RyR2 mutation that renders the channel leaky (RyR2S2808D/S2808D) and a mouse model with a homozygous RyR2 mutation that renders the channel protected against leak (RyR2S2808A/S2808A) [129]. In cardiomyocytes derived from the RyR2S2808D/S2808D mice there was an increased mitochondrial Ca2+ and ROS production, the presence of dysmorphic and malfunctioning mitochondria, a reduction in mitochondrial size and low fusion-to-fission ratio compared with wild-type (WT) and RyR2S2808A/S2808A cardiomyocytes [129]”.

Point 4. Please fix the used terminology. Thus, “failing heart” is not equal to cardiomyopathies. Heart failure needs to be clearly defined and used appropriately, as “cardiac dysfunction” and “post-myocardial infarction” are not equal to “heart failure.”

We thank the reviewer for the elucidation and we amended the text accordingly.

Page 9, lines 419-420: “failing hearts”.

Page 10, line 434: “after postmyocardial infarction”.

Page 11, line 524: “cardiomyopathies”.

Point 5. Several review parts are superficially written and need more specific ROS/Ca-signaling-related details. For instance:

In subsection - 5.4 Drug targeting of mitochondrial Ca2+ and homeostasis:

“Ranolazine is a piperazine derivative selectively inhibiting late INa [132], 2007). It was FDA approved in 2006 for chronic angina [133]. MERLIN-TIMI 36 trials provided us the hypothesis that ranolazine may be particularly beneficial in patients with HF [134].

We fully agree with the Reviewer for the helpful suggestion and we added the modified text in the subsection 5.4:

Page 10, lines 463-472: “In this trial, patients with acute coronary syndrome profited from Ranolazine, particularly when they had elevated levels of brain natriuretic peptide as a marker of HF [138]. Evidence suggested that the mechanisms of action could be the blocking of the late Na+ current that occurs during ischemia, the blocking mitochondrial complex I activity, or by modulating mitochondrial metabolism [139]. In mitochondria isolated from hearts of patients treated with Ranolazine, there was decreased cytochrome c release and mild resistance to the opening of mPTP, when compared with control hearts. Through its late Na+ current blocking action, Ranolazine protected the heart during IR injury, by decreasing the load of cytosolic (c)Ca2+ and mitochondrial (m)Ca2+. The final effect was a reduction in necrosis and apoptosis [139]”.

Point 6. Empagliflozin is a selective inhibitor of the sodium glucose cotransporter 2 (SGLT-2) [135] that is located in the renal early proximal tubule and reabsorbs the majority of filtered glucose.

Empagliflozin decreases renal glucose reabsorption and thus hyperglycaemia [136,137]. A trial in 7020 patients called EMPA-REG OUTCOME showed that empagliflozin reduced several cardiovascular endpoints and in particular, hospitalization for and death from heart failure [138].

Page 10, lines 445-451: “As an example, CGP-37157 is a benzodiazepine compound selectively inhibiting NCLX, the mitochondrial Na+/ Ca2+ exchanger by which mitochondrial Ca2+ is extruded. CGP-37157 suppresses mitochondrial Ca2+ efflux, abolishes NADH oxidation and reduces ROS production in failing cardiomyocytes [110]. Overall, experimental studies performed with CGP-37157 showed the positive effects of mitochondrial NCLX inhibition in animal models of heart failure and in isolated heart cells. However, currently there are no clinical trials to test the effect of these drugs”.

Page 11, lines 477-483: “A clinical trial in 7020 patients, the EMPA-REG OUTCOME study, showed that patients with type 2 diabetes at high cardiovascular risk and treated with Empagliflozin, showed an ameliorated cardiovascular endpoints and reduced death from HF, when the study drug was added to standard care [135,136,137]. Empagliflozin might reduce [Na+]i in cardiac myocytes independently of SGLT inhibition, possibly by interacting with the Na+/H+ exchanger [141]. Thereby, this drug empowered mitochondrial Ca2+ in cardiac myocytes”.

We also discussed in depth other points in section 5.4:

Page 10, lines 453-458: “In heart failure, [Na+]i is increased [132] and during excitation-contraction coupling, it hinders mitochondrial Ca2+ accumulation, since it increases mitochondrial Ca2+ efflux via the NCLX [102]. It has been shown that an elevated late Na+ current (late INa) is required to increase cytosolic Na+. An enhanced late INa has been related to elevated mitochondrial ROS emission. This in turn worsens late INa through Ca2+-/calmodulin-dependent kinase II (CaMKII) oxidation”.

Page 11, lines 486-497: “Despite evidence that mitochondria are crucial for oxidative damage, compartmentalization and the interaction between compartments probably explain why "generic" antioxidants, such as vitamins, failed to target dysfunctional mitochondria and altered Ca2+ levels.

A more targeted and specific approach has been focused on NOXs, which have the biological role to produce ROS for signaling purposes and to transfer electrons across biological membranes to produce O2- [143]. Along with the already well-known lipid-lowering effect of statins in preventing cardiovascular disease, their antioxidant effect through NOX2 inhibition has been demonstrated. Statin treatment provided positive antioxidant effects in vitro in isolated cardiomyocytes and in animal models of HF [144,145]. However, in the CORONA study, comprising 5011 HF patients, statin therapy did not show to reduce cardiovascular death [146]”.

Page 11, lines 498-520: “In January 2021, FDA has approved Vericiguat, a new soluble oral guanylate cyclase stimulator, which enhances the production of cyclic guanosine monophosphate, for the treatment of chronic HF. It is well known that the cardioprotective pathway of soluble nitric oxide guanylate cyclase-cyclic guanosine monophosphate is compromised in patients with HF [148].

Based on data from the SOCRATES-REDUCED and VICTORIA trials in adult patients with chronic HF, this drug was approved for medical treatment. In the VICTORIA trial, the dose of Vericiguat was initially 2.5 mg/day and increased after 2-weeks to 5 mg/day, and then to 10 mg/day. Placebo doses were administered in the same manner. After about 1 year, 90% of patients in both treatment groups were treated with the 10 mg target dose. The median follow-up for the primary endpoint was 11 months. The annualized absolute risk reduction with the drug treatment was ~4.2% over the course of the study [147]. The recommended initial oral dose of Vericiguat was 2.5 mg/day with food intake, and should be doubled approximately every 2 weeks to reach the target maintenance dose of 10 mg/day. Based on the VICTORIA trial, Vericiguat significantly reduced the rate of hospitalization and cardiovascular death attributed to HF. In fact, regardless of patients' atrial fibrillation status, Vericiguat was better in preventing all causes of death, cardiovascular death, hospitalization and HF.

The main advantage of this drug is that it avoids the risk of electrolyte imbalance or renal damage [149]. Moreover, Vericiguat bypasses the many problems of current therapies, such as the gradual decline in effectiveness, drug dose-dependent tolerance, and off-target effects due to a lack of specificity, thus representing a game changer in the treatment of HF, because it has great potential to reduce the severity of the disease [150]”.

Point 7. Conclusions and Future Perspectives:

The many statements are unrelated to the current review and should be removed.

The following statement is misleading - “Interestingly, clinical trials are already underway whose aim is to generate differentiated cardiac cells from cells derived from cardiomyopathy patients and study their molecular mechanisms [153].” Unclear the type of cardiac cells and the clinical trial(s) the authors refer to.

We added more details on the different clinical trials and the corresponding references:

Page 14, lines 612-619: “Tenreiro et al. reported data from trials aimed at generating cardiomyocytes from cells derived from cardiomyopathy patients and study their molecular mechanisms [162]. An important example of this approach is DMDstem (NCT03696628), a trial completed in 2021 and performed in children with genetic cardiomyopathy vs. healthy ones. However, the outcomes of this study are not yet reported in the literature. Another trial, IndivuHeart (NCT02417311) initiated in 2014 and completed in 2019, aims to ensure individualized early risk assessment for heart disease and to make engineered heart tissue technology (hiPSC-EHT) a clinically applicable strategy [163]”.

We thank the reviewer for the helpful suggestions and comments of our manuscript.

Round 2

Reviewer 3 Report

The authors addressed all critical issuers raised in the round 1 review, and the manuscript can be accepted in its present form.